# Psychological distress symptoms among healthcare professionals are significantly influenced by psychosocial work context, Ethiopia: A cross-sectional analysis

Gebisa Guyasa Kabito 👩🦰*, Tesfaye Hambisa Mekonnen

Department of Environmental and Occupational Health and Safety, Institute of Public Health, College of Medicine and Health Sciences, University of Gondar, Gondar, Ethiopia

* gebeguyasa4@gmail.com

## Abstract

### Background

Work-related psychosocial hazards result in numerous adverse effects on physical and psychological health, a reduction in quality of life and a decline in performance capacity to workers. While the condition is widespread across various occupations, healthcare sectors are acknowledged to be at high risk. In developing countries such as Ethiopia, however, the lack of reliable data on psychosocial hazards including psychological distress (PD) often hampers officials from planning for preventive actions. This study assessed the magnitude and influencing factors of PD among healthcare professionals in Gondar city, Ethiopia.

### Methods

We employed a cross-sectional survey of 422 healthcare professionals selected with a stratified random sampling technique between April and March 2019. The symptoms of psychological distress were assessed with a standardized 10 items Kessler Psychological Distress Scale instrument. Bivariate and multivariate analyses were conducted by use of SPSS program version 20. Factors associated with psychological distress symptoms were ascertained at < 0.05 p-value. Adjusted odds ratio (AOR) with a confidence interval (CI) of 95% was used to determine the strength of associations.

### Results

In total, 417questionnaires fully completed and returned with a response rate of 98.8%. Age ranges from 23 to 57 with a mean of 31.00 (± 8.219 SD) years. Work-related psychological distress symptoms in the past 4weeks stood at 44.4% (N = 185) [95%CI (39.8, 49.4)]. Being female worker [AOR: 2.07; 95% CI (1.29, 3.32)], high job demand [(AOR: 1.53; 95% CI (1.10, 2.57)] and low job control [AOR: 2.54; 95% CI (1.60, 4.04)] were significant factors of psychological distress.

**Data Availability Statement:** All relevant data are within the manuscript and its Supporting Information files.

**Funding:** The authors received no specific funding for this work.

**Competing interests:** The authors have declared that no competing interests exist.

**Abbreviations:** AOR, Adjusted Odd Ratio; CI, Confidence Interval; COR, Crude; Odd Ratio; ETB, Ethiopia Birr; IBM SPSS, International Business Machines Corporation Statistical Package for Social Science; PD = Psychological distress; SD, standard deviations.

## Conclusion

This study underscores the level of psychological distress among healthcare professionals is high. The experiences of psychological distress symptoms were significantly influenced by socio-demographic factors including sex and psychosocial job characteristics such as job demand and job control. Efforts to prevent the mental health of workers including psychological distress symptoms need to focus on individual attributes and job contexts.

## Introduction

Newly emerging risks, such as psychosocial hazards in various contemporary workplaces have become major threats to workers, employers and the society. The healthcare sector is one of the vulnerable sectors conceivably linked with exposure to psychosocial and other traditional risks at the same time [1–3]. Psychosocial job characteristics such as a work-related psychosocial distress (PD) in healthcare sectors are important to employees and healthcare systems [3–6]. The consequences of work-related high psychosocial distress in healthcare is much more sizable [4].

First and foremost, the effects are greater for the workers involved, for instance, in terms of in experiencing physical and psychological adverse health effects [7], a decrease in performance capability [8], and a decline in quality of life [5, 9, 10]. In addition, studies have documented that psychological ill health contributes substantially to the overall global disease burden [11, 12], and leads to substantial claims for occupational illnesses [13]. In healthcare, the effects of psychosocial distress can also be explained in relation to the quality of service provided, resulting in low patient satisfaction and higher medical costs [14].

Several epidemiological data have confirmed that the incidence of work-related psychological symptoms varies from 47% in china [15] to 64.6% India [16], 29.9% in Ethiopia [17] to 44.1% Nigeria [18]. The relation of work-related psychological distress and various sociodemographic factors including sex, age, marital status, and experience [19, 20] has been well explored. There is ample evidence that highlights the associations of psychosocial work contents such as high job demand, shift work [21–23], job control [23], and job dissatisfaction [24] with the experiences of psychological distress symptoms. Further, psychological distress symptoms are also affected by life style behavior including alcohol consumption [25], Khat chewing [26], cigarette smoking [27, 28].

In developing countries such as Ethiopia, however, the lack of reliable data on psychosocial hazards including PD often impedes officials from planning for preventive actions [29, 30]. In Ethiopia, the healthcare industry is growing with the subsequent increase in the number of healthcare professionals (HCPs). However, programs to ensure the psychosocial job characteristics which lead to physical and ill mental health are negligible in the country. Addressing the symptoms of PD is imperative not only to protect the health and well-being of workers but also to ensure the quality of care and safety of patients [31]. In this study, we analyzed the prevalence and factors associated with work-related PD among healthcare professionals in Gondar city, Northwest Ethiopia.

## Materials and methods

### Study design, aim, period, and setting

A cross - sectional study design was employed to assess the prevalence and factors of work related PD based on a sample of healthcare professionals in Gondar city from March to April

2019. The city is located in the state of Amhara, 747 km from the capital of Ethiopia, Addis Ababa. There were nine public health facilities in the city; one hospital (Gondar Comprehensive Specialized Hospital) and eight health centers (*Teda*, *Azezo*, *Maraki*, *Ginbot 20*, *Gabriel*, *Poli*, *Woleka*, *and Bilajig*) working for the population of Gondar and nearby. According to the administrative zone health department's plan and program report, there were 2350 health care professionals working in hospitals and health centers at the time of the study.

## Source and study populations

All healthcare professionals in Gondar city administration were the source population. The randomly selected healthcare professionals in the selected public health facilities were our study population. Healthcare professionals who had worked for at least 6 months in the study area prior to the study and who were present and working in the selected health facilities during the study period were included, whereas those who were on sick, annual, maternity, and family leaves were excluded.

## Sample size determination and sampling procedures

The single population proportion formula [32] was used to calculate the sample size with the following assumptions: *p* (proportion of work-related PD0.5since this would yield the maximum sample size), *d* (margin of error = 0.05and *Zα/2* (standard score value for 95% confidence level = 1.96), the minimum sample size (n) was became 422 having considered 10% non-response rate. We used stratified followed by simple random sampling technique to recruit eligible samples. List of healthcare professionals was obtained from Human resource (HR) departments of each healthcare facility. Numbered lists of all healthcare professionals in Gondar city were made and 422 study subjects were randomly selected using the random number generator software. The sample size was proportionally allocated to each health institution based on the number of healthcare professionals working in the respective health facilities. Participants were approached at their respective worksite.

## Data collection tools and variable measurement

Data were collected through a pre-tested and structured self-administered data collection technique. The Kessler 10 Psychological Distress Scale (K10) [33] was used to assess the prevalence of work-related PD. The scale consists of 10 questions that have been asked about the experiences of distress over the last four weeks. Responses were scored on a five-point ordinal scale reflecting how much of the participants had experienced 10 symptoms over the past four weeks, such as "feeling tired for no good reason" and "sad or depressed." Every item on a scale from 1 to 5, ranges in the severity from 'none of the time' to 'all of the time. 'The total K10 score for each participant was calculated by summing all 10 items, which then ranged from a minimum of 10 to a maximum of 50. In line with prior research, we dichotomized the variable, so that participants with scores less than 22 were regarded as not distressed and participants with scores $\geq$ = 22 considered distressed [34, 35]. The K10 has shown reliability and validity in previous research Cronbach's alpha coefficient for the K10 was 0.91 [36]. K10 scale has also previously been validated in Ethiopia [37] and yielded good internal consistency of 0.93, sensitivity of 84.2%, and specificity of 77.8%. On top of this, language validity was performed to make sure that the local language questionnaire is equivalent to the English questionnaire. Therefore, it was reasonable to apply for this study. For this study, Cronbach's α was 0.83. Job content questionnaire (JCQ) [38], a widely validated and reliable measure, was used to measure demands, control, relationships and social support. It is usually used to assess psychosocial factors at work. Job demand was assessed with five items (Cronbach'sα = 0.70). Job

control was measured with seven items; Cronbach's α of the seven-item scale was 0.73. For social support two indicators, with each four items, were used referring to both supervisor support (α = 0.87) and colleague support (α = 0.71), respectively. For each respondent, the mean value was calculated then higher numbers implied higher demands, greater control, and good relationships and support [39]. These measures have shown validity and reliability in previous studies [40, 41].Job satisfaction of participants was assessed using the 10-item generic job satisfaction scale with 32 or above as a cut-off point [42]. Moreover, several demographics were included: age, gender, marital status, work experience, monthly salary, family number, education level and profession. They were also asked about behavioral factors including alcohol use, smoking, and khat chewing.

## Data quality control

We recruited six final year Environmental and Occupational Health and Safety students for data collection and two Occupational Health lecturers, and one psychiatry professionals as supervisors. Two days training was offered for data collectors and supervisors on topics related to research objectives, clarity of questions, the confidentiality of information and consent in the study. The training was given in lecture and discussion ways. The questionnaires were pretested on 15samples that were not included in the final analysis and the relevant modifications were made before the actual data collection was conducted.

## Data management and statistical analysis

The data were checked for completeness and entered into EPI info version 7.1.5.2, and exported to IBM SPSS 20 software for analysis. Frequency distributions, percentages, means, and standard deviations were used for description of the results. Using a binary logistic regression analysis, we fitted each predictor variable in to a bivariate logistic regression model separately to explore associations with the dependent variable (self-reported work-related PD). Explanatory variables with p-value < 0.2 in the bivariate analysis were exported to the multivariable logistic regression model using backward variable selection method. Hosmer and Lemeshow goodness-of-fit test was used to check the model fitness (P>0.05). A multi co-linearity assumption was checked using Variance Inflation Factor (VIF <5). Odds ratios (OR) with 95% confidence intervals (CI) and p -value< 0.05 were applied to establish the significance of associations.

## Ethics approval and consent to participate

Ethical approval was obtained from the ethical committee of the Department of Environmental and Occupational Health & Safety, College of Medicine and Health Sciences, University of Gondar. Participants were informed about the objective of the research by data collectors. Written informed consent was obtained from each study participant. To keep privacy of the information obtained, only aggregate data were used for analysis and interpretations of the results. There were no risks due to participation in this research project. The collected data were used for this research purpose only and kept with complete confidentiality.

## Results

### Socio-demographic and individual characteristics of the respondents

Out of 422 respondents selected, 417 participated and completed the questionnaire giving an overall response rate of 98.8%. The majority, (230, 55.2%) of the participants were males. The mean age was 31.00 years (SD = 8.219; range 23–57 years). Most of the respondents were

married (221, 53%). A high proportion of the respondents had a BSc degree (344, 82.5%), above degree (49, 11.8%), and few (24, 5.8%) of the participants had a diploma as their highest level of qualification. The professional status of the participants was (233, 55.9%) Nurses, (57, 13.7%) Physician, (48, 11.5%) Midwifery, and (79, 18.9%) categorized in others. Most participants (193, 46.3%) had a monthly salary of less than 4466 Ethiopian Birr (ETB) and (85, 20.4%) reported their monthly salary was > 6176 ETB. A total of (367, 88%) participants were non- smokers (Table 1).

## Psychosocial work environment of the study participants

More than half of the respondents (227, 54.4%) reported that they had high job demand. One hundred and sixty-four (39.3%) of the participants indicated they had poor relationships with their staffs, and (239, 57.3%) respondents were satisfied with their current job (profession). The majority of respondents (242, 58%) revealed that they had low control over their job (Table 2).

## Prevalence of work-related psychological distress

The prevalence of work-related PD in the past 4weeks was found to be44.4% (N = 185) [95% CI (39.8–49.4)].

## Factors associated with work-related psychological distress

In a bivariate analysis, predictor variables including sex, educational status, monthly salary, work experience, family size, professional status, habits of cigarette smoking, khat chewing, high job demand, low job control, poor staff relationships, job dissatisfaction, and low perceived organizational support were explored to considerably influence work-related PD.

After controlling for confounders in a multivariable logistic regression analysis, sex, high job demand, and low job control remained to significantly influence the experience of self-reported work-related PD. Accordingly, the probability of having work-related PD was 2.07 times higher among female study participants than males [AOR: 2.07; 95% CI (1.29, 3.32)]. Moreover, the chance of suffering from work-related PD was 1.53 times higher among participants who encountered high job demand than those who reported low job demand [(AOR: 1.53; 95% CI (1.10, 2.57)]. As well, the probable occurrence of work-related PD was increased by two and half among respondents who did not able to control their jobs than among those who have capacity to control their job [AOR: 2.54; 95% CI (1.60, 4.04)] (Table 3).

## Discussion

Because of the changing nature of contemporary workplaces, work-related psychosocial hazards are becoming important areas of study. However, little has been recorded in developing countries like Ethiopia, despite its pervasiveness among workers of various occupations. The aim of this study was to quantify the prevalence of psychological distress and its influencing factors among healthcare professionals in Gondar city, Northwest Ethiopia. This study demonstrated the prevalence of work-related psychological distress among health professionals in the previous4 weeks was 44.4% (N = 185). In Ethiopia, a suboptimal workplace setup coupled with the traumatic events of daily life possibly aggravates the mental health status of employees, including PD. A study in Nigeria reported a comparable finding to the current investigation (44.1) [43]. The possible explanations for this correspondence might be that in developing countries, socio-economic status and the inherent nature of work characteristics in healthcare

**Table 1. Socio-demographic and individual characteristics of respondents among healthcare professionals working in healthcare facilities in Gondar city, Northwest Ethiopia, 2019.**

| Variables(n = 417) | Frequency | Percent (%) |
|---|---|---|
| **Sex** | | |
| Male | 230 | 55.2 |
| Female | 187 | 44.8 |
| **Age in years** | | |
| <30 | 247 | 59.2 |
| > = 30 | 170 | 40.8 |
| **Educational level** | | |
| Diploma | 24 | 5.8 |
| Degree | 344 | 82.5 |
| Above degree | 49 | 11.8 |
| **Marital status** | | |
| Married | 221 | 53.0 |
| Single | 196 | 47.0 |
| **Monthly salary (ETB)** | | |
| <4466 | 193 | 46.3 |
| 4466–6176 | 139 | 33.3 |
| >6176 | 85 | 20.4 |
| **Profession** | | |
| Physician | 57 | 13.7 |
| Nurses | 233 | 55.9 |
| Midwifery | 48 | 11.5 |
| Others[b] | 79 | 18.9 |
| **Work experience** | | |
| ½-5years | 327 | 78.4 |
| Greater than 5years | 90 | 21.6 |
| **Family number** | | |
| Less than 4 | 305 | 73.1 |
| 4 and above | 112 | 26.9 |
| **Religion** | | |
| Orthodox | 284 | 68.1 |
| Muslim | 73 | 17.5 |
| Protestant | 51 | 12.2 |
| Others[a] | 9 | 2.2 |
| **Cigarette smoking** | | |
| Smoker | 50 | 12.0 |
| Non-smoker | 367 | 88.0 |
| **Khat chewing** | | |
| Chewer | 74 | 17.7 |
| Non chewer | 343 | 82.3 |

**Keys:** a = Catholic, Jewish; b = Laboratory, Environmental health, Pharmacist, Physiotherapy, Health officers; ETB = Ethiopian Birr (1$USD = 34.73ETB currency).

facilities such as the lack of adequate resources (human and material), job insecurity, and poorly arranged work conditions invariably likely similar.

Our result is higher than the reports from Sri Lank (21%) [44] and Canada (25.4%) [4]. This might explain that the current study employed a relatively shorter time frame (4 weeks)

**Table 2. Psychosocial working characteristics of respondents among healthcare professionals working in health-care facilities in Gondar city, Northwest Ethiopia, 2019.**

| Variables (N = 417) | Frequency (n) | Percent (%) |
|---|---|---|
| **Job demand** | | |
| Low | 190 | 45.6 |
| High | 227 | 54.4 |
| **Job control** | | |
| High | 175 | 42.0 |
| Low | 242 | 58.0 |
| **Relationships** | | |
| Good | 253 | 60.7 |
| Poor | 164 | 39.3 |
| **Job Satisfaction** | | |
| Dissatisfied | 178 | 42.7 |
| Satisfied | 239 | 57.3 |
| **Social support** | | |
| Low | 229 | 54.9 |
| High | 188 | 45.1 |

**Table 3. Factors associated with work-related psychological distress among healthcare professionals working in healthcare facilities in Gondar city, northwest Ethiopia, 2019.**

| Variable(N = 417) | Work-related PD | | COR(95%CI) | AOR (95%CI) |
|---|---|---|---|---|
| | No | Yes | | |
| **Sex** | | | | |
| Male | 137 | 93 | 1 | 1 |
| Female | 95 | 92 | 1.43 (0.97, 2.11) | 2.07(1.29, 3.32)** |
| **Age** | | | | |
| <30 | 157 | 90 | 0.45 (0.30, 0.67) | 0.42 (0.37, 1.06) |
| > = 30 | 75 | 95 | 1 | 1 |
| **Educational level** | | | | |
| Diploma | 15 | 9 | 0.74(0.27, 2.0) | 0.69 (0.39,1.91) |
| Degree | 190 | 154 | 1.00 (0.55, 1.82) | 0.92 (0.77, 3.92) |
| Above degree | 27 | 22 | 1 | 1 |
| **Marital status** | | | | |
| Married | 115 | 106 | 1.37 (0.93, 2.01) | 1.01 (0.62,1.65) |
| Single | 117 | 79 | 1 | 1 |
| **Monthly salary (ETB)** | | | | |
| <4466 | 134 | 59 | 0.52 (0.31, 0.88) | 0.49 (0.27,1.04) |
| 4466–6176 | 52 | 87 | 1.97 (1.14, 3.41) | 1.56 (0.82,2.97) |
| >6176 | 46 | 39 | 1 | 1 |
| **Work experience(years)** | | | | |
| ½ -5 years | 185 | 142 | 0.84 (0.53, 1.34) | 1.04 (0.55, 2.01) |
| >5years | 47 | 43 | 1 | 1 |
| **Family number** | | | | |
| Less than 4 | 165 | 140 | 1 | 1 |
| 4 and Above | 67 | 45 | 0.79 (0.51, 1.22) | 0.72 (0.62,1.61) |
| **Profession** | | | | |
| Physician | 23 | 34 | 2.55 (1.27, 5.13) | 2.53 (0.98, 5.21) |

(*Continued*)

**Table 3.** (Continued)

| Variable(N = 417) | Work-related PD | | COR(95%CI) | AOR (95%CI) |
|---|---|---|---|---|
| | No | Yes | | |
| Nurses | 138 | 95 | 1.19 (0.70, 2.01) | 1.15 (0.81, 2.52) |
| Midwifery | 21 | 27 | 2.22(1.07, 4.61) | 2.20 (0.96,4.58) |
| Others[b] | 50 | 29 | 1 | 1 |
| **Cigarette smoking** | | | | |
| Smoker | 25 | 25 | 1.29 (0.72, 2.34) | 1.74 (0.89,3.41) |
| Not smoker | 207 | 160 | 1 | 1 |
| **Khat chewing** | | | | |
| chewer | 36 | 38 | 1.41 (0.85, 2.33) | 1.39 (0.95,3.11) |
| Non chewer | 196 | 147 | 1 | 1 |
| **Job demand** | | | | |
| High | 115 | 112 | 1.56 (1.06, 2.31) | 1.53 (1.10, 2.57)*** |
| Low | 117 | 73 | 1 | 1 |
| **Job control** | | | | |
| High | 125 | 50 | 1 | 1 |
| Low | 107 | 135 | 3.15 (2.08, 4.78) | 2.54 (1.60, 4.04)*** |
| **Relation ships** | | | | |
| Good | 134 | 119 | 1 | 1 |
| Poor | 98 | 66 | 0.76 (0.51, 1.13) | 1.33(0.80,2.23) |
| **Job satisfaction** | | | | |
| Satisfied | 121 | 118 | 1 | 1 |
| Dissatisfied | 111 | 67 | 0.62 (0.42, 0. 92) | 0.59 (0.51, 1.08) |
| **Social support** | | | | |
| Low | 138 | 91 | 0.66 (0.45, 0.98) | 0.69 (0.55, 1.39) |
| High | 94 | 94 | 1 | 1 |

**Keys:** **statistically significant at p < 0.01|***statistically significant at p < 0.0001 | b = Laboratory, Environmental health, Pharmacist, Physiotherapy, Health officers.

experience of the workers, which can enhance a recalling capacity of the participants to report recent experiences. However, in this study, symptoms of psychological distress were detected at a lower magnitude than a literature in China (85.5%) [45]. This disparity could be attributed to reporting cultures, definitions, and degrees of perceptions of psychological distress symptoms across the countries.

In accordance with abundant epidemiological data [45–48], psychological distress symptom is markedly influenced by sex of employees in our investigation. Females often tend to shoulder greater responsibilities and accountabilities including caring for children, the elderly, and involving household and social roles in several ways. The combined effects of household/workplace duties eventually predispose females to detrimental psychosocial characteristics such as time pressure and job control. Studies have also elucidated that women are vulnerable to psychological distress because they may perceive discrimination at work, encounter violence, and use emotional oriented coping strategies [47–49]. Furthermore, in developing countries including Ethiopia, the majorities of female workers are engaged in low-level employment status with low incomes, routine, and zero tolerance for deadline, which places females at high risk of developing psychological ill health.

Scholars have widely reported that a low job control is a significant contributor to the symptoms of psychological distress [50, 51]. Our analysis also supported those reports. Employees

with a low job control/autonomy might perceive that job differently, may not be satisfied with, may be stressed/distressed, and may eventually experience ill mental health symptoms in general.

In the multivariable analysis, the incidence of psychological distress was significantly related to a work demand. This finding reproduced a report from another studies [50, 51]. Working in a healthcare setting is indeed a challenging because employees usually need to handle intensive tasks involving work overload, extended working hours, high effort and energy, and limited professionals/lack of support. Even though the stressful nature of working in healthcare facilities is basically similar everywhere, in developing countries like Ethiopia, such challenges are far beyond empirical witness due to a lack of an effective distribution of resources for health.

This study bestows great opportunities for healthcare officials and other stakeholders to account for workplace context when planning for mental health prevention programs. In Ethiopia, the relation of diverse psychosocial job characteristics and ill psychological symptoms has been little researched and therefore the current study inculcates other researchers to shift their interests to this pivotal concern. However, the result needs to be interpreted with cautions when drawing a conclusion. First, the study was a cross-sectional survey of limited healthcare facilities which might preclude the findings from conclusion to other workplaces. Second, because the data obtained were past experiences of workers self-reporting, underreporting cannot be overlooked. To minimize such bias, we restricted only to the recent experiences. Future investigators need to incorporate diverse workplaces to explicitly comprehend the relations of various work characteristics and psychological distress.

## Conclusion

This study underscores the level of psychological distress among healthcare professionals is high. The experiences of psychological distress symptoms were significantly influenced by socio-demographic factors including sex and psychosocial job characteristics such as job demand and job control. Efforts to prevent the mental health of workers including psychological distress symptoms need to focus on individual and job context.

## Supporting information

**S1 Data set. This is data set used in analysis.**
(XLSX)

## Acknowledgments

The authors are prominently thankful for all the data collectors, supervisors, study participants, and the University of Gondar for their creditable assistances to the success of this study.

## Author Contributions

**Conceptualization:** Gebisa Guyasa Kabito, Tesfaye Hambisa Mekonnen.

**Data curation:** Gebisa Guyasa Kabito.

**Formal analysis:** Gebisa Guyasa Kabito, Tesfaye Hambisa Mekonnen.

**Methodology:** Gebisa Guyasa Kabito.

**Supervision:** Gebisa Guyasa Kabito.

**Validation:** Gebisa Guyasa Kabito.

**Visualization:** Gebisa Guyasa Kabito.

**Writing – original draft:** Gebisa Guyasa Kabito.

**Writing – review & editing:** Gebisa Guyasa Kabito, Tesfaye Hambisa Mekonnen.

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
