## [Decision Letter · Decision Letter 0]

3 Jul 2020

PONE-D-20-14581

Psychological distress symptoms among healthcare professionals are significantly influenced by psychosocial work context, Ethiopia: A cross-sectional analysis

PLOS ONE

Dear Dr. Kabito,

Thank you for submitting your manuscript to PLOS ONE. After careful consideration, we feel that it has merit but does not fully meet PLOS ONE’s publication criteria as it currently stands. Therefore, we invite you to submit a revised version of the manuscript that addresses the points raised during the review process.

We look forward to receiving your revised manuscript.

Kind regards,

Amir H. Pakpour, Ph.D.

Academic Editor

PLOS ONE

Journal Requirements:

Reviewers' comments:

Reviewer's Responses to Questions

**Comments to the Author**

1. Is the manuscript technically sound, and do the data support the conclusions?

Reviewer #1: Yes

2. Has the statistical analysis been performed appropriately and rigorously? 

Reviewer #1: Yes

3. Have the authors made all data underlying the findings in their manuscript fully available?

Reviewer #1: Yes

4. Is the manuscript presented in an intelligible fashion and written in standard English?

Reviewer #1: Yes

5. Review Comments to the Author

Reviewer #1: The manuscript is generally well written with clear descriptions in the study purpose. However, some minor points need to be addressed in the revision.

1. Please adhere to the use of defined abbreviations. For example, in line 77, there is a "psychological distress" term. However, the psychological distress has been defined as PD at the very beginning of the manuscript. Line 73 also has the similar problem. I do not point out all of them, the authors should check by themselves throughout the manuscript.

2. Line 118. Here comes a term of "K10". After re-read the manuscript, I can identify that the K10 indicates the Kessler Psychological Distress Scale. However, the authors did not define the K10 when they mention the Kessler Psychological Distress Scale.

3. Line 119. Research cannot be counted. Therefore, the authors should correct "researches" into "research".

4. Line 138. The Data quality control section is the biggest problem in the manuscript. It is unclear why the authors define these actions as data quality control. The section reads like the descriptions of language validity. Moreover, some descriptions are confusing in the section. Specifically, back-translation indicates translating local language back to English. I cannot understand why the authors can back translate the English to local language. Also, did the authors conduct a forward translation? Moreover, ensuring language validity (i.e., make sure that the local language questionnaire is equivalent to the English questionnaire) is not about data quality. The language validity ensures the questionnaire validity.

5. Following the previous comment. If the authors want to present how they control the data, they should inform the readers how they make sure that the data input and data collections have no errors.

6. Table 1. Not everyone is familiar with the Ethiopian Birr. Please provide an exchange rate of the Ethiopian Birr with USD.

7. Also Table 1. I think that the footnote a should go earlier than footnote b.

8. Table 3. I cannot understand why the authors mentioned "Hosmer and Lemeshow test = 0.754 showed that the model fitted well" here. Also, there is a superscript b for Others in the Profession category. However, there is no footnote to explain the meaning of b.

6. PLOS authors have the option to publish the peer review history of their article (what does this mean?). If published, this will include your full peer review and any attached files.

Reviewer #1: No

---

## [Author Response · Author response to Decision Letter 0]

23 Aug 2020

we have uploaded as a separate file of the responses to the editors and reviewers

---

## [Editor Report · Decision Letter 1]

4 Sep 2020

Psychological distress symptoms among healthcare professionals are significantly influenced by psychosocial work context, Ethiopia: A cross-sectional analysis

PONE-D-20-14581R1

Dear Dr. Kabito,

We’re pleased to inform you that your manuscript has been judged scientifically suitable for publication and will be formally accepted for publication once it meets all outstanding technical requirements.

Kind regards,

Amir H. Pakpour, Ph.D.

Academic Editor

PLOS ONE
---

## [Editor Report · Acceptance letter]

10 Sep 2020

PONE-D-20-14581R1 

Psychological distress symptoms among healthcare professionals are significantly influenced by psychosocial work context, Ethiopia: A cross-sectional analysis 

Dear Dr. Kabito:

I'm pleased to inform you that your manuscript has been deemed suitable for publication in PLOS ONE. Congratulations! Your manuscript is now with our production department. 

Kind regards, 

on behalf of

Dr. Amir H. Pakpour 

Academic Editor

PLOS ONE